# Tumor Microenvironmental Cytokines Drive NSCLC Cell Aggressiveness and Drug-Resistance via YAP-Mediated Autophagy

**DOI:** 10.3390/cells12071048

**Published:** 2023-03-30

**Authors:** Paola Matarrese, Rosa Vona, Barbara Ascione, Camilla Cittadini, Annalisa Tocci, Anna Maria Mileo

**Affiliations:** 1Oncology Unit, Center for Gender-Specific Medicine, Italian National Institute of Health, Viale Regina Elena, 299-00161 Rome, Italy; 2Tumor Immunology and Immunotherapy Unit, IRCCS Regina Elena National Cancer Institute, Via Elio Chianesi, 53-00144 Rome, Italy

**Keywords:** non-small cell lung cancer (NSCLC), autophagy, cytokines, microenvironment, cancer progression, epithelial-mesenchymal transition (EMT), invasion, drug-resistance

## Abstract

Dynamic reciprocity between cellular components of the tumor microenvironment and tumor cells occurs primarily through the interaction of soluble signals, i.e., cytokines produced by stromal cells to support cancer initiation and progression by regulating cell survival, differentiation and immune cell functionality, as well as cell migration and death. In the present study, we focused on the analysis of the functional response of non-small cell lung cancer cell lines elicited by the treatment with some crucial stromal factors which, at least in part, mimic the stimulus exerted in vivo on tumor cells by microenvironmental components. Our molecular and functional results highlight the role played by the autophagic machinery in the cellular response in terms of the invasive capacity, stemness and drug resistance of two non-small lung cancer cell lines treated with stromal cytokines, also highlighting the emerging role of the YAP pathway in the mutual and dynamic crosstalk between tumor cells and tumor microenvironment elements. The results of this study provide new insights into the YAP-mediated autophagic mechanism elicited by microenvironmental cytokines on non-small cell lung cancer cell lines and may suggest new potential strategies for future cancer therapeutic interventions.

## 1. Introduction

Modification of the tumor microenvironment (TME) in response to cell transformation is an early event during carcinogenesis and is considered a pre-requisite for tumor progression processes [1,2]. In response to hostile growth conditions, such as tissue hypoxia, deficiency of nutrients, accumulation of waste products and acidity caused by rapid cell proliferation, cancer cells recruit and reprogram stromal and immune cells through a dynamic reciprocity that may promote an immunosuppressive environment able to evade immune surveillance [3,4]. Paracrine communication between the cellular TME components and cancer cells takes place through the interaction of soluble signals, mainly cytokine-related, produced by both tumor and stromal cells. Cytokines, low-molecular-weight protein mediators, are produced by diverse types of cells and commonly mediate intercellular communication within the TME components to support or hinder cancer progression by regulating cell survival, differentiation, immune cell activation, cell migration and cell death. Depending on the specific tumor microenvironment context, cytokines may exert either pro-inflammatory or anti-inflammatory effects, leading to a complex network. Cancer-associated fibroblasts (CAFs), the most abundant and heterogeneous stromal components [5,6], have been widely described as key actors in multiple stages of tumor development via numerous mechanisms, including cytokines secretion and extracellular matrix (ECM) remodeling. Several studies have shown that CAF subtypes exhibit a different immunomodulatory secretion profile that plays a major role in modulating the TME by regulating immune cell recruitment and functions within tumors and participating in the therapy response [7,8]. Based on the relevance of paracrine signaling between cancer cells and CAFs in tumor progression and invasion [9,10,11], we specifically explored the functional responses of non-small cell lung cancer (NSCLC) cell lines treated with different cytokines. Among the processes involved in the crosstalk between the various components of the TME and cancer cells, we focused on the potential role of autophagy in tumor progression [12,13]. Macroautophagy (herein called autophagy) is a highly conserved, tightly regulated physiological process, during which double-membrane vesicles (autophagosomes) sequester dysfunctional cellular components and initiate their lysosome-mediated degradation [12]. In cancer, a controversial role for autophagy has been described, depending on the context, genetic features, stage of the disease and host immunity: autophagy can suppress tumor initiation (cytotoxic autophagy), whereas once the tumor is established, a proficient autophagic process generally supports the survival, proliferation and therapy resistance of cancer cells (cytoprotective autophagy) [13,14,15]. The pro-tumorigenic role of autophagy has been studied and validated by several groups in different human tumors [16,17,18,19,20,21]. Emerging evidence suggests that autophagy in tumor cells can be modulated by various environmental integrated signals, including hypoxia, immune signals, and metabolic, oxidative and mechanical stresses, hence favoring cancer cell invasiveness, tumor progression and drug resistance, at least in the same setting [16,22,23,24]. Interestingly, novel non-canonical autophagy pathways are emerging, demonstrating roles in secretory processes [25,26] and endocytic trafficking [27,28]. The existence of autophagic extracellular delivery mechanisms argues that autophagy is not only a lysosomal degradative process, but may also play important biogenesis roles in the extracellular secretion of bioactive molecules. Autophagy-mediated release of secreted factors may promote immunosuppressive TME properties and, in turn, may promote a cancer invasive phenotype. Furthermore, recent studies [29,30] have shown that autophagy acts as a downstream regulator of the Yes1 Associated Transcriptional Regulator (YAP) pathway. YAP usually acts as a transcriptional coactivator for TEAD transcription factors and is the terminal effector of the Hippo pathway [31,32,33]. Growing evidence, however, suggests that some Hippo-independent signaling pathways upstream of YAP may have a relevant role in cancer progression [34]. Notably, it has been shown that YAP is able to control autophagic flux by regulating both the degradation of autophagosomes and their maturation in autolysosomes [29,35]. In this study, we evaluated the effect of several relevant cytokines known to be secreted by stromal components and involved in the crosstalk between tumor cells and TME signal pathways. Our data highlight the role played by the YAP-related autophagic machinery in the response of two different NSCLC cell lines to treatment with a combination of stromal cytokines in terms of invasive properties, cell stemness and drug resistance, suggesting the emerging pivotal role of YAP in the mutual and dynamic crosstalk between cancer cells and TME components.

## 2. Materials and Methods

### 2.1. Cell Lines and Cell Culture Treatments

The human NSCLC cell lines A549 and Calu-1 were purchased from American Type Culture Collection (ATCC, Rockville, MD, USA) and routinely cultured in RPMI 1640 medium (Invitrogen Corporation, Carlsbad, CA, USA) supplemented with 10% FBS and antibiotics. Recombinant human TGF-β was obtained from R&D systems (240-B-010/CF) and IL-6, IL-8 and CCXL-16 from PeproTech, Inc. (London, UK). Before each treatment, i.e., the combined cytokine (CytoMix), cisplatin (CDDP) or CytoMix + CDDP, cells were cultured for 16 h in FBS-free RPMI 1640 medium. The total concentration of the cytokines mixture used for the stimulation experiments was 10 ng/mL (2.5 ng/mL for each of the following cytokines: TGF-β, IL-6, IL-8 and CCXL-16). For cell death induction, cells were treated with 100 µM (A549) or 200 µM (Calu-1) CDDP (Sigma-Aldrich Inc., St. Louis, MO, USA) for 24 h in the presence or absence of CytoMix.

### 2.2. Cell Migration Assay

Cell migration was examined with a scratch assay, according to Liang et al. [36], and performed as described [37]. Approximately 2.5 × 10^5^ cells were seeded in 35 mm Ø dishes. When the cells reached confluence, the dishes were scratched with a sterile 20 µL pipette tip. The cells were then washed with phosphate-buffered saline (PBS), photographed at T0 and treated with 10 ng/mL of CytoMix. The migration of cells toward the wound closure of the same region after 48 h was monitored, and images (10 fields/dish) were acquired using a digital camera system coupled with an inverted microscope (IX-71 Olympus Corporation, Tokyo, Japan). Repopulation by migrating cells of the wound region was then analyzed and quantified using the ImageJ v1.48 software.

### 2.3. Cell Invasion Assay

The invasion ability of the cells was evaluated to quantify cell migration through 8 μm-pore-sized polycarbonate membrane Matrigel-coated transwell inserts (Corning Inc., Corning, NY, USA). The bottom chamber included a medium (0.5 mL) containing FBS. A549 and Calu-1 were seeded in 0.5 mL of medium without FBS into the upper chamber and incubated at 37 °C in a humidified atmosphere containing 5% CO_2_ for 48 h. After this time, cells on the upper surface were mechanically removed with a cotton swab, and the membranes were then washed, fixed with 4% paraformaldehyde and stained using Hoechst 33258 (Sigma-Aldrich Inc., St. Louis, MO, USA). The invasion capability was quantified by counting with a fluorescence microscope the cells that had migrated to the lower side of the membrane. In addition, the cells that, having passed through the filter, had adhered to the bottom of the well containing the insert were counted through a phase-contrast microscope. With both approaches, comparable results were obtained. All experiments were performed in triplicate. In each experiment, at least five fields were counted.

### 2.4. Immunoblot Analysis

The whole-cell extract was obtained in RIPA lysis buffer in the presence of standard protease and phosphatase inhibitors. The equal protein content of cell lysates was resolved on polyacrilamide gel (Invitrogen, Carlsbad, CA, USA), electro-transferred to PVDF membranes and incubated with the following specific antibodies: anti-LC3 (Novus Biologicals, Centennial, CO, USA; 1:1000); anti-p62 (Cell Signaling Technology (CST), Danvers, MA, USA; 1:1000); anti-β-Actin (MP Biomedicals, Santa Ana, CA, USA; 1:10,000); anti-GAPDH (Santa Cruz Biotechnology Inc., Dallas, TX, USA; 1:2000); anti-YAP (Santa Cruz Biotechnology, Inc.; 1:1000); anti-TBC1D2 (Thermo Fisher Scientific, Walthman, MA, USA; 1:1000); anti-AKT (CST; 1:1000); anti-pAKT (S473) (CST; 1:1000); anti-pAKT (T308) (CST; 1:1000); anti-E-cadherin (Santa Cruz Biotechnology; 1:500); and anti-Vimentin (Santa Cruz Biotechnology; 1:500). After membranes washing, immune complexes were detected with Horse Radish Peroxidase-conjugated species-specific secondary antibodies (Jackson Laboratory, Bar Harbor, ME, USA). The membranes were developed using ECL detection reagents (Millipore Corporation, Billerica, MA, USA). Reactive bands were detected with the ChemiDocMP system (Bio-Rad Laboratories Inc., Hercules, CA, USA) or UVITEC imaging system (UVITEC, Cambridge, UK). The arbitrary units obtained were used to calculate the relative increase/decrease of bands. To ensure the presence of equal amounts of protein, the membranes were re-probed with anti-GAPDH or anti-β-Actin. Protein expression levels were quantified with densitometric analysis using the NIH ImageJ v1.48 software.

### 2.5. Fluorescence Microscopy

For mitochondria labeling, living cells were incubated with 5 µM MitoTracker red (Invitrogen) for 45 min at 37 °C. After this time, the cells were washed in PBS, fixed in 4% paraformaldehyde and counterstained with Hoechst 33258. For YAP, TBC1D2, LC3 and p62 staining, the cells were fixed with 4% paraformaldehyde, permeabilized by 0.5% (*v/v*) Triton X-100 and incubated for 1 h at 4 °C with specific primary antibodies: LC3 (Novus Biologicals; 1:100); p62 (Cell Signaling Technology; 1:100); YAP (Santa Cruz Biotechnology; 1:100); and TBC1D2 (Thermo Fisher Scientific; 1:100). AlexaFluor 488-conjugated anti-mouse IgG and AlexaFluor 594-conjugated anti-mouse IgG (both Invitrogen) were used as secondary antibodies. After washing, all samples were counterstained with Hoechst 33258 and then mounted in fluorescence mounting medium (Dako, Glostrup, Denmark). Images were acquired with intensified video microscopy (IVM) with an Olympus fluorescence microscope (Olympus Corporation, Milan, Italy) equipped with CoolLed pE-300-W (CoolLED Ltd., Andover, UK).

### 2.6. siRNAs Transfection

A549 and Calu-1 NSCLC cells were seeded in 60 mm-diameter plates (3 × 10^5^ cells per plate), and after 24 h, were transfected with 20 nM Silencer Select pre-designed and validated small interfering RNAs for YAP1 (Ambion, Thermo Fisher Scientific, cat. 4392420) or with a non-silencing siRNA (Ambion, Fisher Scientific, cat. 4390843) using Lipofectamine RNAiMAX transfection reagent (Invitrogen), according to the manufacturer’s protocol. BLOCK-iT Alexa Fluor Red Fluorescent Control (Invitrogen) was used to evaluate the cells’ transfection efficiency. After 24 h of siRNA-mediated gene silencing, the cells were treated with the cytokine mixture for the performance of molecular and/or functional investigations, as described above. In every RNA interference experiment, the knockdown level of YAP was quantified from western blot densitometric analysis.

### 2.7. Flow Cytometry Analyses

Cell Viability. Control and treated cells were incubated with 1 μm Calcein-AM (Thermo Fisher Scientific) at 37 °C for 30 min. In live cells, non-fluorescent Calcein-AM is converted to a green fluorescent dye (green-fluorescence-positive cells), whereas dead cells with compromised cell membranes do not retain Calcein and thus do not display green fluorescence (green-fluorescence-negative cells).

Apoptosis. Control and treated cells were stained using a fluorescein isothiocyanate (FITC)-conjugated Annexin V (AV) and propidium iodide (PI) detection kit (Marine Biological Laboratory, Woods Hole, MA, USA). This assay enables identifying both early (AV positive/PI negative) and late apoptotic cells (AV/PI double positive) or necrotic cells (single PI positive).

Autophagy. Evaluation of autophagy was performed by staining cells with a Cyto-ID Autophagy Detection Kit (Enzo Life Sciences, Farmingdale, NY, USA) optimized for detection of autophagy in live cells by flow cytometry [38].

GSH. The intracellular glutathione level (GSH) was detected by staining living cells with monochlorobimane (MBC, Molecular Probes) staining, as previously described [39]. 

Expression of stem cell markers. Cell surface expression of CD34, CD45, CD133, CD146 and CD309 (VEGFR2) was quantified with flow cytometry after staining living cells for 30 min on ice with monoclonal phycoerythrin (PE)-conjugated antibody anti-human CD309 or with monoclonal anti-human allophycocyanin (APC)-conjugated antibodies specific to CD34, CD45, CD133 and CD146 (all BioLegend, London, UK). As negative controls, we used mouse IgG-PE and mouse IgG-APC. After washings in cold PBS, samples were immediately analyzed using a cytometer. 

Fluorescence Resonance Energy Transfer (FRET). We applied FRET analysis with flow cytometry to study the molecular association of YAP and TBC1D2 in untreated and CytoMix-treated cells. Briefly, cells were fixed and permeabilized, as described above, for immunofluorescence and labeled with antibodies tagged with donor (PE) or acceptor (Cy5) dyes. The following primary and secondary antibodies were used: anti-YAP (Santa Cruz Biotechnology), anti-TBC1D2 (Thermo Fisher Scientific), PE-labeled anti-mouse (Sigma-Aldrich) and Cy5-labeled anti-rabbit (AbCam, Cambridge, MA, USA). YAP protein was detected in the FL2 channel (PE, donor), TBC1D2 in FL4 (Cy5, acceptor) and FRET in the FL3 channel. Quantification of protein−protein interactions was obtained by calculating the FRET efficiency (FE) by using the following Riemann algorithm [40]: FE = [FL3DA − FL2DA/a − FL4DA/b]/FL3DA, in which A is the acceptor and D the donor, and where a = FL2D/FL3D and b = FL4A/FL3A.

### 2.8. Data Analysis and Statistics

For flow cytometry studies, samples were acquired with a FACScalibur cytometer (BD Biosciences Inc., San Diego, CA, USA) equipped with a 488 nm Argon laser and with a 635 nm red diode laser and analyzed using CellQuest software (BD Biosciences). For GSH quantification, samples were acquired with a LRS II cytometer (Becton & Dickinson, San Jose, CA, USA) equipped with a 488 nm Argon laser and a UVB laser and analyzed with DIVA software (Becton & Dickinson). At least 20,000 events were acquired for each sample. 

The quantification of proteins and the GSH level was expressed as the median fluorescence intensity. The quantification of cell toxicity was reported as the % of dead cells. Collected data analysis was carried out with ANOVA one-way testing using GraphPad Prism 5 software (GraphPad, San Diego, CA, USA). All data were verified in at least three independent experiments and are reported as means ± standard deviation (SD). A *p* value less than 0.05 was considered as statistically significant; (*) *p* ≤ 0.05 and (**) *p* ≤ 0.01.

## 3. Results

The secretome of CAF subtypes is known to cause cancer progression by both enhancing pro-tumorigenic induction and maintaining an immunosuppressive environment [41]. Based on the emerging relevance of tumor–stromal cells crosstalk and the characterization data of the CAF secretion profile [11], we focused on cellular responses induced by cytokines and chemokines, such as IL-6 and IL-8, which promote cancer stemness, as well as TGF-β and CXCL-16, which are involved in metastatic cancer cells spreading [42,43]. To partially mimic the impact of factors secreted by the microenvironment on tumor progression, we evaluated the functional effects of the aforementioned cytokines (2.5 ng/mL for each of the following cytokines, TGF-β, IL-6, IL-8 and CCXL-16, for a total of 10 ng/mL) in two NSCL cancer cell lines, as well as A549 and Calu-1 arising from an adenocarcinoma and an epidermoid carcinoma, respectively, as they represent the most common histotypes of NSCLC.

### 3.1. CytoMix Promotes Cell Motility and Invasiveness in NSCLC Cells

Tumor progression requires increased invasive capacity, dissemination and metastasis, stemness acquisition and drug resistance. Thus, we firstly evaluated the impact of the cytokine mixture (hereafter referred to as CytoMix) on the cell migration capacity with the wound healing assay. 

Data were collected by photographing the lesions at the times 0, 24 and 48 h following the induction of the lesion. Control cells incubated in serum-free medium (CTR) showed a lower motility and reduced ability to repopulate the wound area than cells treated with CytoMix, as shown in the representative images and histograms for A549 (Figure 1A) and Calu-1 (Figure 1B). In A549 cells, the CytoMix treatment significantly increased the repopulation of the wounded area after 48 h (i.e., about 13% wound reduction in CTR vs. about 30% in cells treated, *p* < 0.01). Comparable results were achieved in the Calu-1 cell line (about 20% wound reduction in CTR vs. about 45% in cells treated, *p* < 0.01). On the contrary, 24 h were not sufficient to induce any appreciable increase in cell motility, both in the untreated control and in CytoMix-treated cells (T0 vs. T24, *p* > 0.05). As a positive control, we used cells incubated in medium with FBS. In the presence of 5% FBS, cell migration was already appreciable after 24 h, and after 48 h, practically the entire wounded area had been repopulated (Figure 1A,B, left panels).

Cell invasiveness was then evaluated by using Matrigel invasion chambers (Figure 2). The invasion capability was calculated both by counting the Hoechst 33258-stained cells in the lower part of the membrane after removing cells from the upper side of the membrane (bottom micrographs) and by counting the cells adherent at the bottom of the well containing the insert (upper micrographs), as specified in the Section 2. Both approaches yielded overlapping results. A549 (Figure 2A) and Calu-1 (Figure 2B) cells treated with CytoMix showed a significant increase of the ability to cross the Matrigel-coated filter to the underlying compartment containing FBS medium, as compared with untreated control cells (Figure 2A,B, middle panels). 

Given the importance of the role of the mitochondria intracellular distribution during migration and invasion [44], we analyzed with fluorescence microscopy the mitochondria network in untreated and CytoMix-treated cells stained with MitoTracker Red and Hoechst 33258 (Figure 2A,B, right panels). The exposure of cells for 24 h to CytoMix induced a significant change in the cell morphology and a redistribution of mitochondria. In accordance with literature data, both in A549 and Calu-1 cells, we observed a re-localization of mitochondria from a perinuclear to the peripheric area of the cell (Figure 2A,B, see arrows in bottom right micrographs).

### 3.2. CytoMix Promotes Epithelial–Mesenchymal Transition (EMT) and Increases Cell Surface Expression of Stem Cell Markers and GSH Content in NSCLC Cells

Since it has been documented that cell invasiveness during cancer progression may be critically dependent on the acquisition of (epithelial–mesenchymal transition) EMT features [45], we aimed to evaluate the changes of the epithelial vs. mesenchymal cell phenotype of NSCLC cells treated or not with CytoMix. We quantified with western blot analysis the cellular content of Vimentin and E-cadherin (mesenchymal and epithelial marker, respectively) in A549 (Figure 3A) and Calu-1 (Figure 3B). In agreement with the functional data reported above, we found that cytokine treatment induced an increase of Vimentin and a substantial decrease of E-cadherin expression, suggesting a CytoMix-induced activation of EMT in both cell lines (Figure 3A,B, upper panels).

Recently, EMT activation has been linked to the acquisition of a stem cell phenotype, proposing EMT as a critical regulator of the CSC state [46,47]. Both EMT activation and cell stemness have a crucial role in invasion and metastasis processes, the onset of chemoresistance and the immune escape phenomenon [48].

Surface expression of some relevant stem cell markers was subsequently investigated with flow cytometry in the control and CytoMix-treated cells. In particular, we considered CD133, one of the most well-characterized stemness marker [49]; CD106, a marker of the mesenchymal phenotype; CD34, conventionally considered as a marker for hematopoietic progenitors, but also expressed on some cancer stem cell populations [48]; CD146, whose overexpression has been linked to the acquisition of drug resistance and the stem phenotype in most cancers, including lung cancer [50]; CD45, whose expression has been associated with the stem phenotype and drug resistance in colon cancer cells; and CD309, (VEGFR2) involved in EMT and angiogenesis in NSCLC [51].

Semiquantitative cytofluorimetric analyses of the cell surface expression of these markers revealed changes in the expression of CD133 and CD146, and to a lesser extent, of VEGFR2. Calu-1 (Figure 3B, bottom panels) had a significantly higher surface expression of CD133 and CD146 than A549 in basal conditions. After treatment with CytoMix, we observed increased levels of CD133 and CD146 only in A549 cells, while no significant changes were observed in Calu-1 (Figure 3A,B, bottom panels). On the contrary, treatment with CytoMix induced a significant increase of VEGFR2 in Calu-1 cells only.

Gluthatione (GSH) metabolism, in addition to EMT and stem cell-like properties, contributes to the acquisition of drug resistance in cancer cells [52,53]. Thus, we evaluated the intracellular content of GSH with flow cytometry. We observed that: (i) untreated Calu-1 had significantly higher GSH content than the A549 one; (ii) CytoMix treatment significantly increased the intracellular GSH content in both cell lines, but to a greater extent in A549 (Figure 3A,B, right bottom panels). These findings suggest that high levels of GSH, by increasing the antioxidant capacity, contribute to the acquisition of resistance to oxidative stress in CytoMix-treated NSCLC cells.

### 3.3. CytoMix Reduces the CDDP Toxicity in NSCLC Cells

Recent studies have highlighted the crucial role played by TME through various mechanisms in the acquisition of resistance to various chemotherapeutics in general and in particular to Cisplatin (CDDP) [54,55,56,57,58]. Since CDDP is commonly used in the treatment of NSCLC [59,60], we tested the effect of CytoMix on the cytotoxicity induced by CDDP in A549 and Calu-1 cells. Flow cytometric assay with Calcein-AM, in association with Annexin V/Propidium iodide analysis, allowed us to quantify cytotoxicity and apoptosis in the same experimental setup. The data reported in Figure 4 indicate that the CytoMix treatment significantly reduced the toxicity of CDDP in both cell lines, thus suggesting its protective effect. The Annexin V positivity test also clearly indicated that CDDP induced an apoptotic death. Furthermore, the greater resistance of Calu-1 cells compared to A549 appears evident (Figure 4A,B), with a reduced susceptibility to CDDP evident even at double the drug concentration (200 µM vs. 100 µM administered to A549).

### 3.4. CytoMix Induces Autophagy Process via YAP Signaling in NSCLC Cells

We, therefore, clarify whether CytoMix could induce autophagy in A549 and Calu-1 cells. For this purpose, the expression level of standard markers of autophagy was analyzed after 24 h of treatment. Although slight autophagy is known to be rapidly activated under stressful conditions, including serum deprivation [61], we used a serum-free culture medium for cytokine treatments to avoid unwanted and poorly controlled stimulation by serum components and, therefore, to improve the reproducibility of the cellular response.

As shown in Figure 5, western blotting analysis of LC3I/II and p62 revealed a clear induction of autophagy in response to CytoMix in both NSCLC cell lines. CytoMix treatment significantly reduced p62 by concomitantly increasing the LC3II amount in both A549 (Figure 5A) and Calu-1 (Figure 5B) cells. These results were confirmed by a semiquantitative cytofluorimetric evaluation of autophagy performed by using the Cyto-ID Autophagy Detection Kit. In addition, fluorescence microscopy analysis showed that the co-localization of p62 with LC3 (yellow areas) observed in untreated cells significantly increased 24 h after CytoMix administration (Figure 5A,B). 

Increasing evidence points to the mammalian target of rapamycin complexes 1/2 (mTORC 1/2) as negative modulators of autophagy. Furthermore, since a complex crosstalk between mTOR pathways and YAP activity has been demonstrated [62], we sought to determine the involvement of mTORC1 and/or mTORC2 in autophagy induction mediated by CytoMix. To gain mechanistic insight into the autophagy process, we analyzed the expression level of Akt^Ser473^, Akt^Thr308^ and total AKT in A549 and Calu-1 treated with CytoMix (24 h). As shown in Figure 5, cytokine treatment, as well as inducing autophagy, specifically reduced AKT phosphorylation at Ser473, but not at Thr308 (Figure 5A, A549 cells; Figure 5B, Calu-1 cells, middle panels). Considering that mTORC2 is the main activator of AKT phosphorylation at Ser473, we propose a crucial role of the mTORC2 axis downregulation in NSCL cancer cells’ autophagic process.

Based on the growing evidence on the crosstalk between YAP and the autophagic process [29,63], we hypothesized that YAP might have a critical role in NSCLC progression by modulating autophagosome biogenesis and its maturation. As shown in Figure 5, YAP was highly expressed in both A549 and Calu-1 cells in basal conditions. CytoMix treatment induced a marked reduction of the YAP expression level, thus suggesting that YAP could be partly degraded through the autophagy process [64]. Recent studies suggested that YAP may modulate autophagic flux by transcriptional regulation of TBC1D2 [30] and, in a transcription-independent manner, by interacting with TBC1D2 in the cytoplasm [65].

Western blot analyses revealed that TBC1D2 content was significantly increased in CytoMix-treated A549 and Calu-1 cells (Figure 6A,B, respectively). Quantitative FRET also demonstrated an increase of TBC1D2/YAP association after CytoMix administration in both cell lines (Figure 6A,B, bottom panels). In representative immunofluorescence images of cells labeled for YAP (green) and TBC1D2 (red), large areas of co-localization of these two proteins (yellow) could be observed, especially in cells treated with CytoMix (Figure 6A,B, upper left panels).

### 3.5. YAP siRNA Inhibits CytoMix-Induced Cell Autophagy and Invasion in NSCLC Cells

Small interfering RNA (siRNA) was employed for the ablation of YAP and its function to deeply characterize the role of YAP in the autophagic process and establish whether YAP is an effector and/or a target of CytoMix-induced autophagy. Western blot analyses of siRNA-treated cells revealed that YAP expression was significantly reduced compared to control cells (Figure 7A, A549 cells; Figure 7B, Calu-1 cells). The transfection efficiency, confirmed by the PE fluorescent-labeled siRNA positive control, was approximately 70% in A549 and 55% in Calu-1 cells (left panels). The data show that YAP silencing affected autophagy induction after CytoMix administration (upper right panels). Functionally, our analysis revealed a significant reduction of CytoMix-induced cell invasion in YAP siRNA-transfected cells as compared to non-silenced cells (Figure 7A,B bottom panels, A549 and Calu-1, respectively), suggesting that YAP expression is relevant for autophagy induction.

## 4. Discussion

NSCLC represents the leading cause of cancer death due to the high percentage of recurrence and its lofty metastatic capability [66,67]. Despite the advancement in knowledge on molecular and cellular mechanisms underlying the oncogenic transformation, the ability of lung cancer cells to metastasize and to acquire chemoresistance remain relatively poorly understood. 

Cancer progression is traditionally viewed as a cascade of events, by which a (pre)-neoplastic cell initially acquires the ability to grow locally without restriction, to undergo EMT, to reach a metastatic niche via the circulation and to form metastatic lesions, as well as acquiring resistance to anticancer drugs [68]. However, cancer cells alone are not sufficient to sustain this complex sequence of events. While the altered expression of oncogenes and tumor suppressor genes in tumor cells constitute the “initial insult”, growing evidence indicates that the TME forms the tumor niche that co-evolves with neoplastic cells and sustains tumorigenesis through various mechanisms [4,69,70]. Conventionally, the focus on the study of tumor initiation, progression and metastasis has been placed on neoplastic cells. This suggests that alterations in oncogenes and tumor suppressor genes result in aberrations of growth signaling pathways that ultimately lead to the uncontrolled proliferation of cancer cells. However, even though these pathways are critical for the metastatic transformation of cells, the unrestrained proliferation of cancer cells cannot be linked only to mutations in the cancer cells themselves. Starting from these considerations, it has been proposed that a dynamic and complex crosstalk between tumor cells and microenvironment components is crucial, both during carcinogenesis and tumor progression [69]. For that reason, the TME in lung tumors is actually recognized as a target-rich environment for the development of novel anticancer therapies. 

CAFs, the major stroma components, predominantly arise from tissue resident fibroblasts and are affected by the stimuli of the tumor. Tumor-derived stimuli that can transform fibroblasts into CAFs are factors secreted by the tumor and the immune infiltrate, such as transforming growth factor β (TGF-β) family ligands, lysophosphatidic acid (LPA), fibroblast growth factor (FGF), platelet-derived growth factor (PDGF), interleukin-1 (IL-1), IL-6 and granulin [71,72,73]. Furthermore, fibroblasts can also physically interact with cancer cells and be activated through crucial signaling pathways [74,75,76]. Moreover, cancer-derived exosomes, delivering cytokines, such as TGF-β [77] or non-coding RNA [78,79], can reprogram fibroblasts into CAFs. 

While it is widely recognized that in vitro cell models cannot recapitulate every aspect of the tumor microenvironment, they nevertheless allow for the mechanistic exploration of the individual contributions of various components of the TME on tumor cell behavior. In the present paper, we explored the functional behavior of two NSCLC cell lines, A549 and Calu-1, following treatment with a combination of soluble factors, i.e., IL-6, IL-8, TGF-β and CXCL-16, known to be secreted by CAFs I [9,10,11,38,39,80,81] to recreate, at least in part, the stimuli exerted in vivo on tumor cells by a microenvironment permissive for cancer growth and progression.

It is well known that cancer cells undergoing EMT lose the epithelial adhesion characteristics and the apical-basal polarity, acquiring a more aggressive mesenchymal phenotype characterized by increased migratory and invasive properties [46]. In this study, we provide evidence that in the NSCLC cells, CytoMix treatment significantly increases cell motility and cell invasion properties, as demonstrated with the scratch assay and cell invasion assay. Furthermore, growing evidence suggests that mitochondrial localization in tumor cells can be reprogrammed based on intracellular and extracellular signals, resulting in the shift of tumor cells from a proliferative to an invasive phenotype, and vice versa [82,83,84]. Starting from these considerations, we evaluated the mitochondrial localization, both in A549 and in Calu-1 treated with CytoMix, demonstrating that the cell morphological shift towards a mesenchymal phenotype is accompanied by the abundance of mitochondria at the cell periphery. This morphological and functional shift, suggestive of the acquisition of a more aggressive tumor cell phenotype, has been associated in our experimental model with changes in the expression level of classic EMT markers, such as loss of epithelial E-cadherin and upregulation of mesenchymal Vimentin. 

In the last decade, the notion of cancer stem-like cells (CSCs) has emerged and assumed increasing scientific relevance. This subset of undifferentiated cancer cells within the tumor bulk is characterized by pluripotency and self-renewal capability, and by the ability to generate heterogeneous cell lines and to promote tumor progression [85,86]. Increasing evidence shows that CSCs can cause tumor recurrence and metastasis, as they have a crucial role also in drug resistance acquisition [87,88]. The increased expression of cell surface markers CD133 and CD146 in NSCLC cells treated with cytokine mixture suggested that treatment promotes cell stemness, thus contributing to the cancer progression process. 

Although targeting and immunotherapy have revolutionized cancer care, chemotherapy-based treatment remains the gold standard for patients with NSCLC. Nonetheless, resistance to chemotherapy causes treatment failure and eventually death.

Regarding the drug resistance of chemotherapeutic agents, the complex role played by reactive oxygen species (ROS) has been extensively described [89,90]. Notably, it has been highlighted that ROS detoxification mechanisms can also promote drug detoxification, reducing the damage and lethality of these agents on the cancer cells, thus contributing to chemoresistance [91]. Accordingly, reduced glutathione (GSH) metabolism has been repeatedly called into question in this regard, and enzymes, such as Glutathione peroxidase 2, have been found upregulated in patients with NSCLC resistant to CDDP [92,93]. Starting from this consideration, we evaluated the CDDP chemosensitivity of A549 and Calu-1 cells treated with CytoMix. The data reported demonstrate that cells treated with the cytokine mixture show reduced CDDP cytotoxicity and more resistance to death. The increase in the GSH intracellular amount induced by cytokine treatment, as well as the higher GSH content observed in the Calu-1 cell line, the most resistant to CDDP treatment, would seem to reinforce the above observations on the dysregulation of NSCLC cells redox homeostasis CytoMix-mediated.

Soluble factors produced by CAFs (i.e., cytokines, chemokines, growth factors, etc.) can confer cancer cell resistance and may activate survival pathways, such as autophagy [47]. It has now been widely established that autophagy plays dual roles in cancer cell survival and death in the context of tumor initiation and development [94]. However, growing evidence shows that the autophagy process acts by, among others, promoting tumor cell survival and growth in extremely stressful conditions, such as hypoxia and nutrient deprivation [95]. A high autophagy level, which helps cancer cells overcome these environmental stresses, has often been observed in metastatic tumors [96,97]. The extent to which autophagy contributes to metastatic processes remains largely unknown [98]. In our experimental model, the exposure of A549 and Calu-1 cell lines to the combination of the previously mentioned cytokines induced the activation of the autophagic process, the concurrent mTORC2-AKT pathway downregulation and modulation of YAP expression. Numerous studies have shown a complex crosstalk between the mTOR kinase signaling pathway and YAP, suggesting that YAP and mTOR proteins, by regulating each other, may contribute to cancer progression [99,100]. Furthermore, it has been reported that although YAP controls autophagic flux by regulating the autophagosomes’ turnover, transcriptional YAP activity is also essential for the maturation of autophagosomes into autolysosomes [30,35]. In addition, YAP is also an autophagy substrate, which is degraded into autolysosomes [64]. Collectively, these findings suggest that there is a complex crosstalk between YAP and the autophagic machinery. In our experimental setting, we found that YAP expression is critical for the autophagosome biogenesis, as demonstrated by the effects of YAP silencing on cell invasion assays, while autophagy negatively regulated YAP activity through its degradation in the late stages of the catabolic process. Since our data obtained on NSCLC cell lines indicated a proteasome-independent down-expression of YAP (data not shown), we hypothesized, in agreement with literature data [64], that YAP was degraded in autolysosomes. Moreover, recent studies suggest that YAP can also modulate the autophagic flux, either through transcriptional regulation [30] or by directly interacting in the cytoplasm with TBC1D2 [63]. TBC1D2 is also an effector of Rac1 [101], a small GTPase that regulates cytoskeletal remodeling, migration and adhesion events [102]. Consistent with these data, we documented a significant increase in TBC1D2 and its enhanced interaction with YAP following cytokine treatment. Furthermore, the inhibition of cytokine-induced effects, in terms of autophagy and invasive potential, through the knockdown of YAP by siRNA, spoke for a key role of YAP-related autophagy in the acquisition of pro-invasive cellular features in both A549 and Calu-1 cell lines.

In summary, the exposure to the aforementioned stromal-related cytokines induced a functional switch in the two NSCLC cell lines indicative of the acquisition of a more aggressive tumor phenotype. The increased migratory and invasive capacity, as well as the acquisition of CDDP drug resistance, were associated with a spatial redistribution of mitochondria, with the EMT phenomenon and with an increase both in the membrane expression of stemness markers and in the intracellular GSH content. 

## 5. Conclusions

Our findings reveal the existence of an autophagy/YAP crosstalk regulating the multistep and finely tuned NSCL cancer progression process induced by TME cytokines. It is conceivable that targeting the YAP-autophagy axis could provide new and effective therapeutic tools focused on the TME activities that drive tumor cell reprogramming towards more aggressive phenotypes.

## Figures and Tables

**Figure 1 cells-12-01048-f001:**
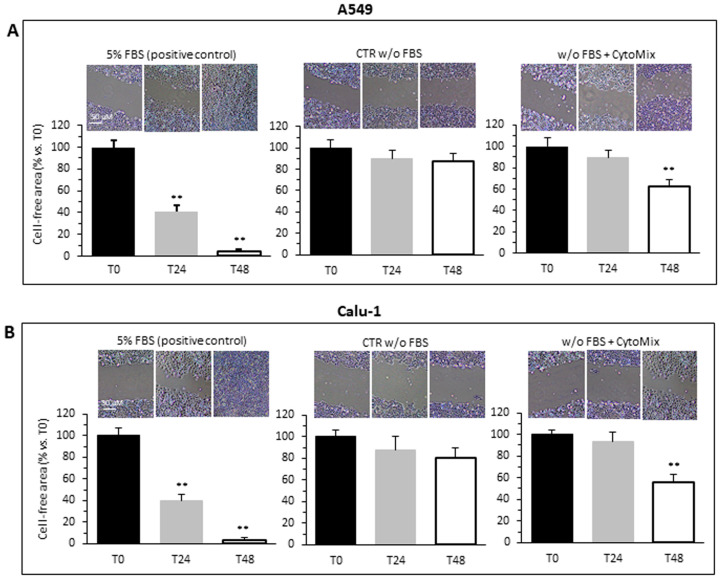
Cells migration. Migration test performed by scratch assay on two different NSCC cancer cell lines: (**A**) A549 and (**B**) Calu-1 grown in medium with FBS (left), medium w/o FBS (central) or medium w/o FBS plus CytoMix (right). Upper panels: representative phase-contrast microscopy images of three independent experiments for both control and CytoMix-treated cells. In the bar graphs (bottom panels), quantification of the area reduction at 24 h and 48 h was represented as the percentage of wounded area at T0. Representative phase-contrast images (20 × objective) of four independent experiments are shown. The wound healing area was analyzed by using ImageJ software (NIH) and the corresponding data expressed as mean ± SD. (**) *p* < 0.01 vs. T0 (one-way ANOVA).

**Figure 2 cells-12-01048-f002:**
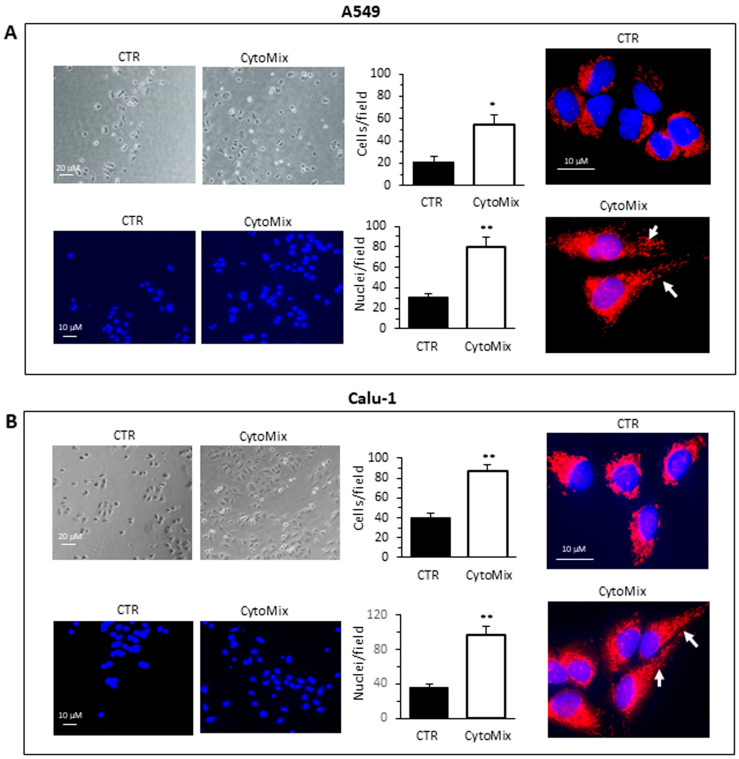
Cells invasion. Invasion assays performed on the NSCC cell lines (**A**) A549 and (**B**) Calu-1 using transwell culture inserts (8.0 µm pore size) coated with Matrigel. Upper left panels: representative phase-contrast images (20 × objective) show cells that have invaded through the insert after 48 h and are at the bottom of the well containing the insert. Bottom left panels: representative fluorescence micrographs of Hoechst 33258-stained cells show cells migrating in the lower part of the membrane. Results reported in bar graphs represent the mean ± SD of three different experiments. (*) *p* < 0.05 and (**) *p* < 0.01 vs. CTR (one-way ANOVA). In the right panels, fluorescence micrographs representative of three independent experiments for control (upper) and CytoMix-treated cells (bottom) show cells stained with MitoTracker red, to label the mitochondrial network, and counterstained with Hoechst 33258.

**Figure 3 cells-12-01048-f003:**
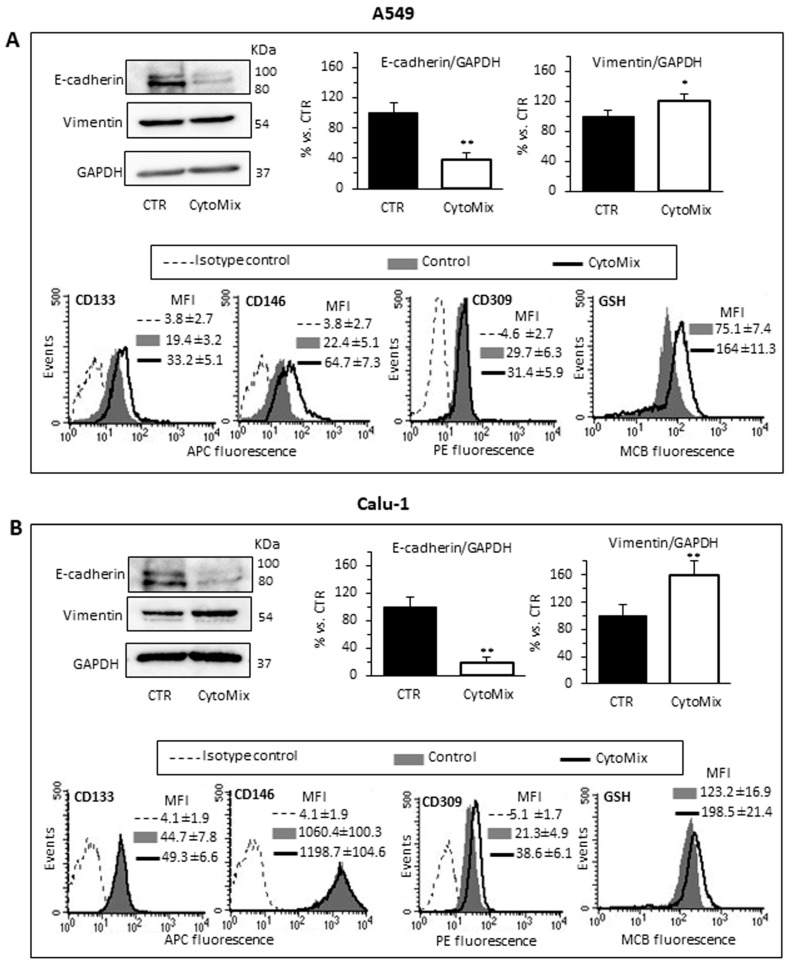
EMT, stem cell markers and GSH. Analyses of EMT and stem cell markers in (**A**) A549 and (**B**) Calu-1 cells untreated (CTR) or treated for 24 h with CytoMix. Upper left panels: representative western blot analysis of E-cadherin, Vimentin and GAPDH used as loading controls. Bar graphs show densitometry analyses of E-cadherin and Vimentin. Each protein is normalized to GAPDH. Data are expressed as mean ± SD of three independent experiments. (*) *p* ≤ 0.05 and (**) *p* ≤ 0.01 vs. CTR (one-way ANOVA). Bottom panels: semiquantitative flow cytofluorimetry analyses of plasma membrane expression level of CD133, CD146 and CD309 (VEGFR2) and GSH intracellular level in a representative experiment among three. Numbers represent the mean of the median fluorescence intensity (MFI) values ± SD obtained in three independent experiments.

**Figure 4 cells-12-01048-f004:**
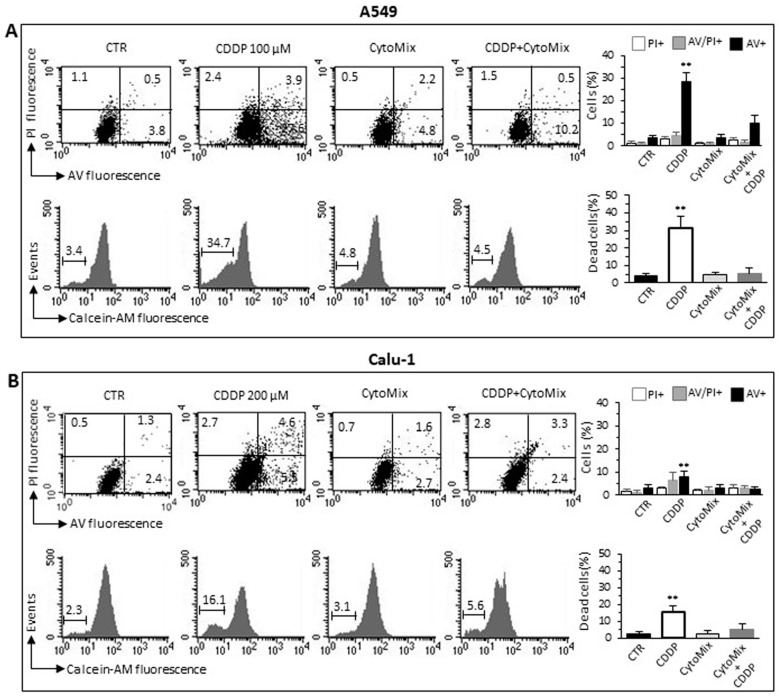
CDDP-induced cell death. Cell death analyses in (**A**) A549 and (**B**) Calu-1 cells treated for 24 h with CDDP (100 µM and 200 µM for A549 and Calu-1, respectively), CytoMix (10 ng/mL) or CDDP plus CytoMix. Upper panels: biparametric flow cytometry analysis of apoptosis. Numbers represent the percentage of apoptotic cells: annexin V/Propidium iodide double-positive (upper right quadrant, grey columns), annexin V single-positive (bottom right quadrant, black columns) or propidium iodide single-positive (necrotic cells, upper left quadrant, white columns). Bottom panels: flow cytometric analysis after cell staining with Calcein-AM (which is retained in the cytoplasm of live cells). Data reported in the bar graph indicate the percentage of Calcein-negative cells (dead cells). Results obtained in a representative experiment are shown. Bar graphs represent the mean ± SD of three different experiments. (**) *p* < 0.01 vs. CTR (one-way ANOVA).

**Figure 5 cells-12-01048-f005:**
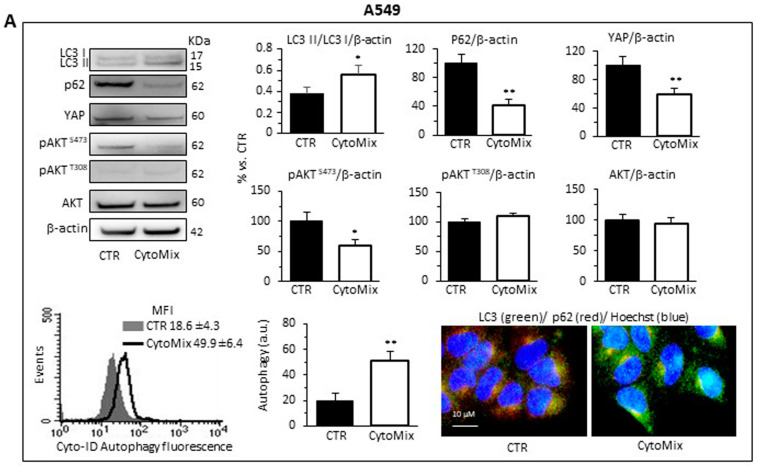
Autophagy analysis and YAP activity. Upper and middle panels: western blot analysis of autophagy by monitoring LC3, p62, YAP, pAKT^S473^, pAKT^T308^ and total AKT proteins, using GAPDH or β-actin as a loading control, in (**A**) A549 and (**B**) Calu-1 cells untreated (CTR) or treated for 24 h with CytoMix. On the left is shown a representative western blot experiment. Bar graphs show densitometric analyses of each protein normalized to GAPDH or β-actin. Data are expressed as mean ± SD of three independent experiments. Bottom panels: flow cytometry curve obtained in a representative experiment performed using Cyto-ID Autophagy Detection Kit. The nearby bar graph shows data obtained in three independent experiments expressed as mean ± SD of the median fluorescence intensity (MFI). (*) *p* ≤ 0.05 and (**) *p* ≤ 0.01 vs. CTR (one-way ANOVA). The immunofluorescence micrographs on the right show representative images of untreated (CTR) or CytoMix-treated cells stained for p62 (red) and LC3 (green) and counterstained with Hoechst (blue). Yellow areas indicate colocalization of LC3 and p62.

**Figure 6 cells-12-01048-f006:**
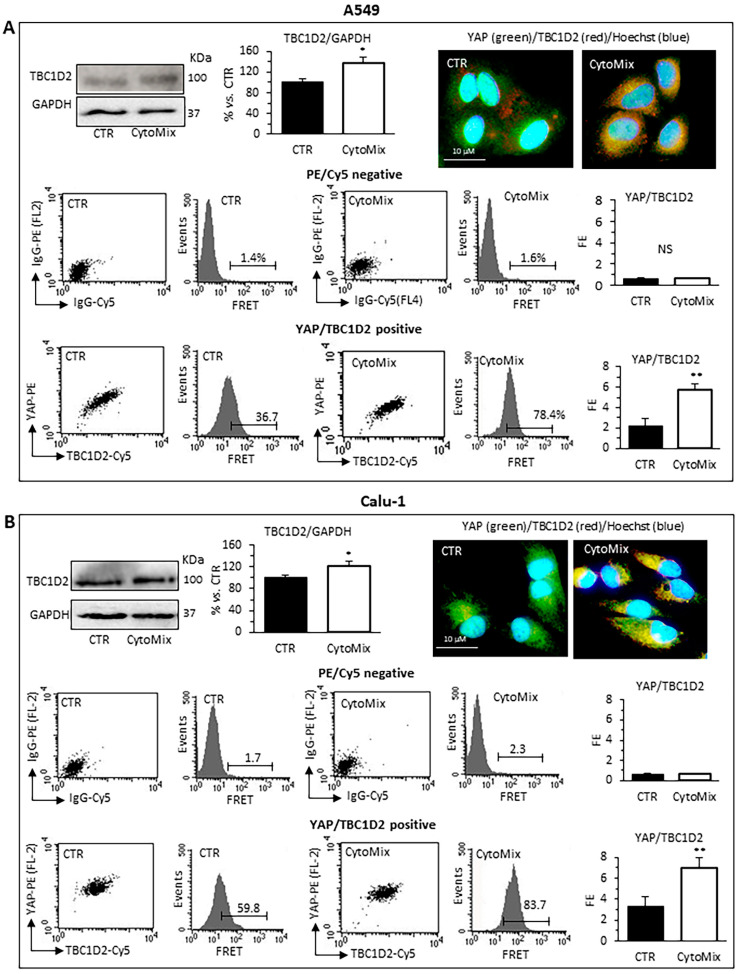
TBC1D2 analysis. Upper panels: western blot analysis of TBC1D2, using GAPDH as a loading control, in (**A**) A549 and (**B**) Calu-1 cells untreated (CTR) or treated for 24 h with CytoMix. On the left is shown a representative western blot experiment. Bar graph shows densitometric analysis of protein normalized to GAPDH. Data are expressed as mean ± SD of three independent experiments. The immunofluorescence micrographs on the right show representative images of untreated (CTR) or CytoMix-treated cells stained for TBC1D2 (red) and YAP (green) and counterstained with Hoechst 33258 (blue). Yellow areas indicate colocalization of TBC1D2 and YAP. Middle and bottom panels: quantitative evaluation of TBC1D2-YAP association by FRET technique as revealed by flow cytometry analysis. Numbers, in flow cytometry curves panels indicate the percentage of FL3-positive events (FRET channel) obtained in one experiment representative of three. Bar graphs at left show the FRET efficiency, calculated according to the Riemann’s algorithm, of TBC1D2 and YAP molecular association. Data are reported as mean ± SD from three independent experiments. (*) *p* < 0.05 and (**) *p* < 0.01 vs. CTR (one-way ANOVA).

**Figure 7 cells-12-01048-f007:**
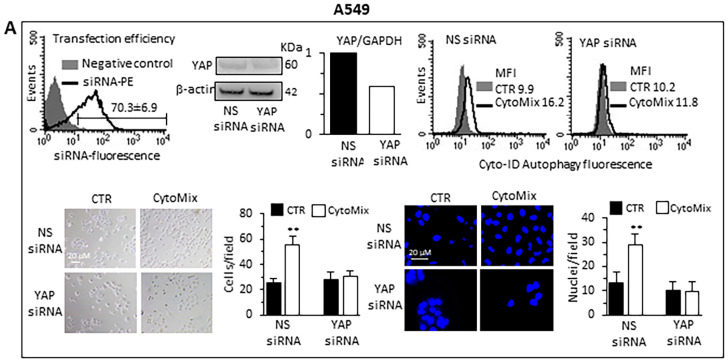
Knockdown of YAP by siRNA. (**A**) A549 and (**B**) Calu-1 cells were transfected with Alexa Fluor Red Fluorescent Control siRNA to verify transfection efficiency, with specific siRNA to knock down YAP expression, or with a non-silencing siRNA (CTR siRNA) as reference control. Upper panels: on the left are shown representative cytometric curves to evaluate transfection efficiency. Number represents the mean percentage ± SD of the cells positive to red fluorescence obtained in two independent experiments performed in triplicate. Central and right panels show autophagy quantification by flow cytometry in cells transfected with YAP-silencing siRNA (central) or non-silencing siRNA (right) left untreated (CTR) or treated with CytoMix obtained in a representative experiment using Cyto-ID Autophagy Detection Kit. Numbers represent median fluorescence intensity values (MFI). Bottom panels: analyses of the invasion capability in transfected cells untreated (CTR) or treated with CytoMix. Representative phase-contrast images show cells that have invaded through the insert after 48 h and are at the bottom of the well containing the insert; representative fluorescence micrographs of Hoechst 33258-stained cells show cells migrating in the lower part of the membrane. Results reported in bar graphs represent the mean ± SD of three different experiments. (**) *p* < 0.01 vs. CTR (one-way ANOVA).

## Data Availability

The data supporting the findings are available.

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
