# Peer review of "Tumor Microenvironmental Cytokines Drive NSCLC Cell Aggressiveness and Drug-Resistance via YAP-Mediated Autophagy"

_cells, 2023, doi:10.3390/cells12071048_

Round 1

Reviewer 1 Report

The aggressiveness and drug-resistance of lung cancer cells were investigated in the present study. Several suggestions and concerns are as below.

1. In Abstract, the abbreviation "TME" appeared for the first time without full substantive. Additionally, "EMT" in Keywords did not be shown in Abstract.

2. Combined cytokine is critical treatment for experimental cells, while the composition was not clarified. In 2.1, the authors merely demonstrated that IL-6 and IL-8 were puchased; and sussessively explained that "the total concentration of cytokines mixture ... was 10 ng/mL (2.5 ng/mL of each cytokines)." The above illustration seemed not very clear enough and was inconsistent.

3. As mentioned above, the influence of several cytokines that altered cancer progression were discussed in Discuss section. The relationship between cytokines mixture used in the present study is pivotal.

4. The insight of the present study is potential for future cancer therapeutic interventions.

Author Response

Reviewer 1

Comments and Suggestions for Authors

The aggressiveness and drug-resistance of lung cancer cells were investigated in the present study. Several suggestions and concerns are as below.

  1. In Abstract, the abbreviation "TME" appeared for the first time without full substantive. Additionally, "EMT" in Keywords did not be shown in Abstract.

Authors: We thank the reviewer for the accuracy with which he read our paper and believe her/his input has been invaluable to improve the quality of this manuscript. Following her/his suggestion, we have removed all abbreviations from the Abstract, and in the keywords we have inserted the full name corresponding to EMT.

  1. Combined cytokine is critical treatment for experimental cells, while the composition was not clarified. In 2.1, the authors merely demonstrated that IL-6 and IL-8 were puchased; and sussessively explained that "the total concentration of cytokines mixture ... was 10 ng/mL (2.5 ng/mL of each cytokines)." The above illustration seemed not very clear enough and was

Autors: We apologize for the inaccuracy. In paragraph 2.1 of the revised version of the manuscript we specified the total concentration of cytokinemixture (CytoMix, 10 ng/ml) administered to the cells as well as the quantities of individual cytokines (2.5 ng/ml for each of the following cytokines: TGF-β, IL-6, IL-8 and CCXL-16).

  1. As mentioned above, the influence of several cytokines that altered cancer progression were discussed in Discuss section. The relationship between cytokines mixture used in the present study is pivotal.

Authors: We fully agree with the reviewer on the key role played in the crosstalk between tumor cells and TME by secreted cytokines, in general, and by those considered in our study. The choice of the cytokine mixture used for the NSCLC cells treatment was inspired by both consolidated literature data and our unpublished data obtained by the secretome analysis of CAF populations isolated from lung cancer patients.

  1. The insight of the present study is potential for future cancer therapeutic interventions.

Authors: We thank the reviewer for her/his positive comments on our paper.

Reviewer 2 Report

Title: Tumor microenvironmental cytokines drive NSCLC cell aggressiveness and drug resistance via YAP-mediated autophagy

The authors presented elegant research to shed the light on the role of the YAP pathway in TME by regulating the autophagy mechanism. they evaluated the effect of certain cytokines secreted by stromal components and involved in the crosstalk between tumor cells and the TME signal pathway, on YAP-related autophagic machinery. their data suggested that YAP might have a critical role in NSCLC progression by modulating autophagosome biogenesis and its maturation.

This research suggests that targeting the YAP-autophagy axis could provide new and effective therapeutic tools focused on the TME activities that drive tumor cell reprogramming towards 598

more aggressive phenotypes

A few edits needed to be addressed are:

Fig 4 A and B . the % occ eels bar graph is not clear. It’s better to have each bar separated not overlap for better presentation.

Fig 5 A and B. since the amount of autophagy activation depends on the ratio between LC3I and LC3II, it better shows the ratio of LC3II over LC3I first and then normalizes it to B-actin

Author Response

Reviewer 2

Comments and Suggestions for Authors

Title: Tumor microenvironmental cytokines drive NSCLC cell aggressiveness and drug resistance via YAP-mediated autophagy

The authors presented elegant research to shed the light on the role of the YAP pathway in TME by regulating the autophagy mechanism. they evaluated the effect of certain cytokines secreted by stromal components and involved in the crosstalk between tumor cells and the TME signal pathway, on YAP-related autophagic machinery. their data suggested that YAP might have a critical role in NSCLC progression by modulating autophagosome biogenesis and its maturation.

This research suggests that targeting the YAP-autophagy axis could provide new and effective therapeutic tools focused on the TME activities that drive tumor cell reprogramming towards more aggressive phenotypes.

Authors: We thank the reviewer for the positive and encouraging comments and for her/his constructive suggestions. We amended it to incorporate all the suggestions provided by the reviewer. We would like to thank the reviewer again for taking the time to review our manuscript.

A few edits needed to be addressed are:

Fig 4 A and B the % occ eels bar graph is not clear. It’s better to have each bar separated not overlap for better presentation.

Authors: As suggested, we have replaced the overlapping bar graphs with new graphs with separate bars to make the results clearer and their interpretation more straightforward.

Fig 5 A and B. since the amount of autophagy activation depends on the ratio between LC3I and LC3II, it better shows the ratio of LC3II over LC3I first and then normalizes it to B-actin

Authors: Thanks for this suggestion. We fully agree that the quantification of autophagy essentially depends on the ratio of LC3I to LC3II. Therefore, we did a new densitometric analysis by first calculating this ratio and subsequently normalizing it with housekeeping genes ( b-actin and GAPDH respectively).  Figure 5 A and 5B were modified accordingly.